# Interplay between the cell envelope and mobile genetic elements shapes gene flow in populations of the nosocomial pathogen *Klebsiella pneumoniae*

**Matthieu Haudiquet**[1,2]*, **Amandine Buffet**[1], **Olaya Rendueles**[1©], **Eduardo P. C. Rocha**[1©]

**1** Microbial Evolutionary Genomics, Institut Pasteur, CNRS, UMR3525, Paris, France, **2** Ecole Doctoral FIRE–Programme Bettencourt, CRI, Paris, France

© These authors contributed equally to this work.

\* matthieu.haudiquet@pasteur.fr

**Data Availability Statement:** All individual quantitative observations (S1 to S18 Dataset) that underlie the data summarized in the figures and results of the manuscript are available at: https://

## Abstract

Mobile genetic elements (MGEs) drive genetic transfers between bacteria using mechanisms that require a physical interaction with the cellular envelope. In the high-priority multidrug-resistant nosocomial pathogens (ESKAPE), the first point of contact between the cell and virions or conjugative pili is the capsule. While the capsule can be a barrier to MGEs, it also evolves rapidly by horizontal gene transfer (HGT). Here, we aim at understanding this apparent contradiction by studying the covariation between the repertoire of capsule genes and MGEs in approximately 4,000 genomes of *Klebsiella pneumoniae* (Kpn). We show that capsules drive phage-mediated gene flow between closely related serotypes. Such serotype-specific phage predation also explains the frequent inactivation of capsule genes, observed in more than 3% of the genomes. Inactivation is strongly epistatic, recapitulating the capsule biosynthetic pathway. We show that conjugative plasmids are acquired at higher rates in natural isolates lacking a functional capsular locus and confirmed experimentally this result in capsule mutants. This suggests that capsule inactivation by phage pressure facilitates its subsequent reacquisition by conjugation. Accordingly, capsule reacquisition leaves long recombination tracts around the capsular locus. The loss and regain process rewires gene flow toward other lineages whenever it leads to serotype swaps. Such changes happen preferentially between chemically related serotypes, hinting that the fitness of serotype-swapped strains depends on the host genetic background. These results enlighten the bases of trade-offs between the evolution of virulence and multidrug resistance and caution that some alternatives to antibiotics by selecting for capsule inactivation may facilitate the acquisition of antibiotic resistance genes (ARGs).

figshare.com/projects/Supplementary_Datasets/
114459 The 3980 genomes assemblies analyzed in
this study are publicly available from the RefSeq
database (accession numbers in S1 Dataset) All the
genomes of strains used for experimental evolution
and conjugation assays are publicly available from
ENA (accession numbers in S1 Table).

**Funding:** This work was supported by an ANR
JCJC (Agence national de recherche) grant [ANR
18 CE12 0001 01 ENCAPSULATION] awarded to O.
R. The laboratory is funded by a Laboratoire
d'Excellence 'Integrative Biology of Emerging
Infectious Diseases' grant [ANR-10-LABX-62-
IBEID], the INCEPTION program[PIA/ANR-16-
CONV-0005], and the FRM [EQU201903007835].
M.H. has received funding from the FIRE Doctoral
School (Centre de Recherche Interdisciplinaire,
programme Bettencourt) to attend conferences.
The funders had no role in the study design, data
collection and interpretation, or the decision to
submit the work for publication.

**Competing interests:** The authors have declared
that no competing interests exist.

**Abbreviations:** ARG, antibiotic resistance gene;
BBH, bidirectional best hits; CFU, colony-forming
unit; CLT, capsular locus type; DAP,
diaminopimelic acid; GT, glycosyl transferases;
GTA, gene transfer agent; HGT, horizontal gene
transfer; ICE, integrative conjugative element; Kpn,
*Klebsiella pneumoniae*; LB, Luria–Bertani; MGE,
mobile genetic element; MPF, mating pair
formation; ST, sequence type; WGA, whole-
genome alignment; wGRR, gene repertoire
relatedness weighted by sequence identity; WT,
wild type.

# Introduction

Mobile genetic elements (MGE) drive horizontal gene transfer (HGT) between bacteria, which may result in the acquisition of virulence factors and antibiotic resistance genes (ARGs) [1,2]. DNA can be exchanged between cells via virions or conjugative systems [3,4]. Virions attach to specific cell receptors to inject their DNA into the cell, which restricts their host range [5]. When replicating, bacteriophages (henceforth phages) may package bacterial DNA and transfer it across cells (transduction). Additionally, temperate phages may integrate into the bacterial genome as prophages, eventually changing the host phenotype [4]. In contrast, DNA transfer by conjugation involves mating pair formation (MPF) between a donor and a recipient cell [6]. Even if phages and conjugative elements use very different mechanisms of DNA transport, both depend crucially on interactions with the cell envelope of the recipient bacterium. Hence, changes in the bacterial cell envelope may affect their rates of transfer.

*Klebsiella pneumoniae* (Kpn) is a gut commensal that has become a major threat to public health [7,8] because it is acquiring MGEs encoding ARGs and virulence factors at a fast pace [2,9]. This propensity is much higher in epidemic nosocomial multidrug-resistant lineages than in hypervirulent strains producing infections in the community [10]. Kpn is a particularly interesting model system to study the interplay between HGT and the cell envelope because it is covered by a nearly ubiquitous Group I (or Wzx/Wzy dependent) polysaccharide capsular structure [11,12], which is the first point of contact with incoming MGEs. Similar capsule loci are present across the bacterial phylogeny [13]. There is one single capsule locus in Kpn [14], located between *galF* and *ugd*, which has increased rate of recombination and HGT compared to the rest of the genome [11,15,16]. This locus contains conserved genes encoding the proteins necessary for the assembly and export of the capsule, which is a multistep biosynthesis pathway. These conserved genes flank a highly variable region encoding enzymes that determine the oligosaccharide combination, linkage, and modification (and thus the serotype) [17]. The biochemical determination of the serotype has not been done for the bacteria corresponding to the most recently sequenced genomes. But the genetic content of the capsule locus has been shown to be a very good predictor of the capsule serotype. Such predictions are called capsular locus types (CLTs) to distinguish them from experimentally determined serotypes [17]. Here, we will use CLT to refer to the genomic predictions and serotype to mention the capsule type. There are more than 140 genetically distinct CLTs, of which 76 have well-characterized chemical structures and are referred to as serotypes [17]. Kpn capsules can extend well beyond the outer membrane, up to 420 nm, which is 140 times the average size of the peptidoglycan layer [18]. They enhance cellular survival to bacteriocins, immune response, and antibiotics [19–21], being a major virulence factor of the species. Intriguingly, the multidrug-resistant lineages of Kpn exhibit higher capsular diversity than the virulent ones, which are almost exclusively of the serotype K1 and K2 [10].

By its size, the capsule can hide phage receptors and block phage infection [22]. Since most Kpn are capsulated, many of its virulent phages evolved to overcome the capsule barrier by encoding serotype-specific depolymerases in their tail proteins [23,24]. For the same reason, phages have evolved to use the capsule for initial adherence before attaching to the primary cell receptor. Hence, instead of being hampered by the capsule, many Kpn phages have become dependent on it [25,26]. This means that the capsule may affect the rates of HGT positively or negatively depending on how it enables or blocks phage infection. Furthermore, intense phage predation may select for capsule swap or inactivation, because this renders bacteria resistant to serotype-specific phages. Serotype swaps may allow cells to escape phages to which they were previously sensitive, but they may also expose them to new infectious phages. In contrast, capsule inactivation can confer pan-resistance to capsule-dependent phages [25].

Regarding the effect of capsules on conjugation, very little is known, except that it is less efficient between a few different serotypes of *Haemophilus influenzae* [27]. The interplay between MGEs (phages and conjugative elements) and the capsule has the potential to strongly impact Kpn evolution in terms of both virulence and antibiotic resistance because of the latter's association with specific serotypes and MGEs.

HGT in Kpn is thought to take place by conjugation or in virions, since it is not part of the known naturally transformable bacteria [28]. Hence, the capsule needs MGEs to vary by HGT, but may block the acquisition of the very same MGEs. Moreover, capsulated species are associated with higher rates of HGT [29]. There is thus the need to understand the capsule's precise impact on gene flow and how the latter affects capsule evolution. Here, we leverage a very large number of genomes of Kpn to investigate these questions using computational analyses that are complemented with experimental data. As a result, we propose a model of capsule evolution involving loss and regain of function. This model explains how the interplay of the capsules with different MGEs can either lower, increase, or rewire gene flow depending on the way capsules affect their mechanisms of transfer.

## Results

### Gene flow is higher within than between serotype groups

We reasoned that if MGEs are specifically adapted to serotypes, then genetic exchanges should be more frequent between bacteria of similar serotypes. We used Kaptive [17] to predict the CLT in 3,980 genomes of Kpn. Around 92% of the isolates could be classed with good confidence level. They include 108 of the 140 previously described CLTs of *Klebsiella* spp. The pangenome of the species includes 82,730 gene families, which is 16 times the average genome. It contains 1,431 single copy gene families present in more than 99% of the genomes that were used to infer a robust rooted phylogenetic tree of the species (average ultra-fast bootstrap of 98%, Fig 1A). Rarefaction curves suggest that we have extensively sampled the genetic diversity of Kpn genomes, its CLTs, plasmids, and prophages (Fig 1B). We then inferred the gains and losses of each gene family of the pangenome using PastML and focused on gene gains in the terminal branches of the species tree predicted to have maintained the same CLT from the node to the tip (91% of branches). This means that we can associate each of these terminal branches with one single serotype. We found significantly more genes acquired (co-gained) in parallel by different isolates having the same CLT than expected by simulations assuming random distribution in the phylogeny (1.95×, Z-test $p < 0.0001$, Fig 1C). This suggests that Kpn exhibits more frequent within-serotype than between-serotype genetic exchanges.

Given the tropism of Kpn phages to specific serotypes, we wished to clarify if phages contribute to the excess of intra-CLTs genetic exchanges. Since transduction events cannot be identified unambiguously from the genome sequences, we searched for prophage acquisition events, i.e., for the transfer of temperate phages from one bacterial genome to another. We found that 97% of the strains were lysogens, with 86% being poly-lysogens, in line with our previous results in a much smaller dataset [25]. In total, 9,886 prophages were identified in the genomes. Their genes account for 16,319 families (19.5%) of the species pangenome (Fig 1B). We then measured the gene repertoire relatedness weighted by sequence identity (wGRR) between all pairs of prophages. The wGRR is a measure of genetic similarity that amounts to 0 if there are no homologs between two genomes, and one if all genes of the smaller genome have a homolog with 100% identity in the other genome. This matrix was clustered, resulting in 2,995 prophage families whose history of vertical and horizontal transmissions was inferred using the species phylogenetic tree (see "Prophage detection"). We found 3,269 independent infection events and kept one prophage for each of them. We found that pairs of independently

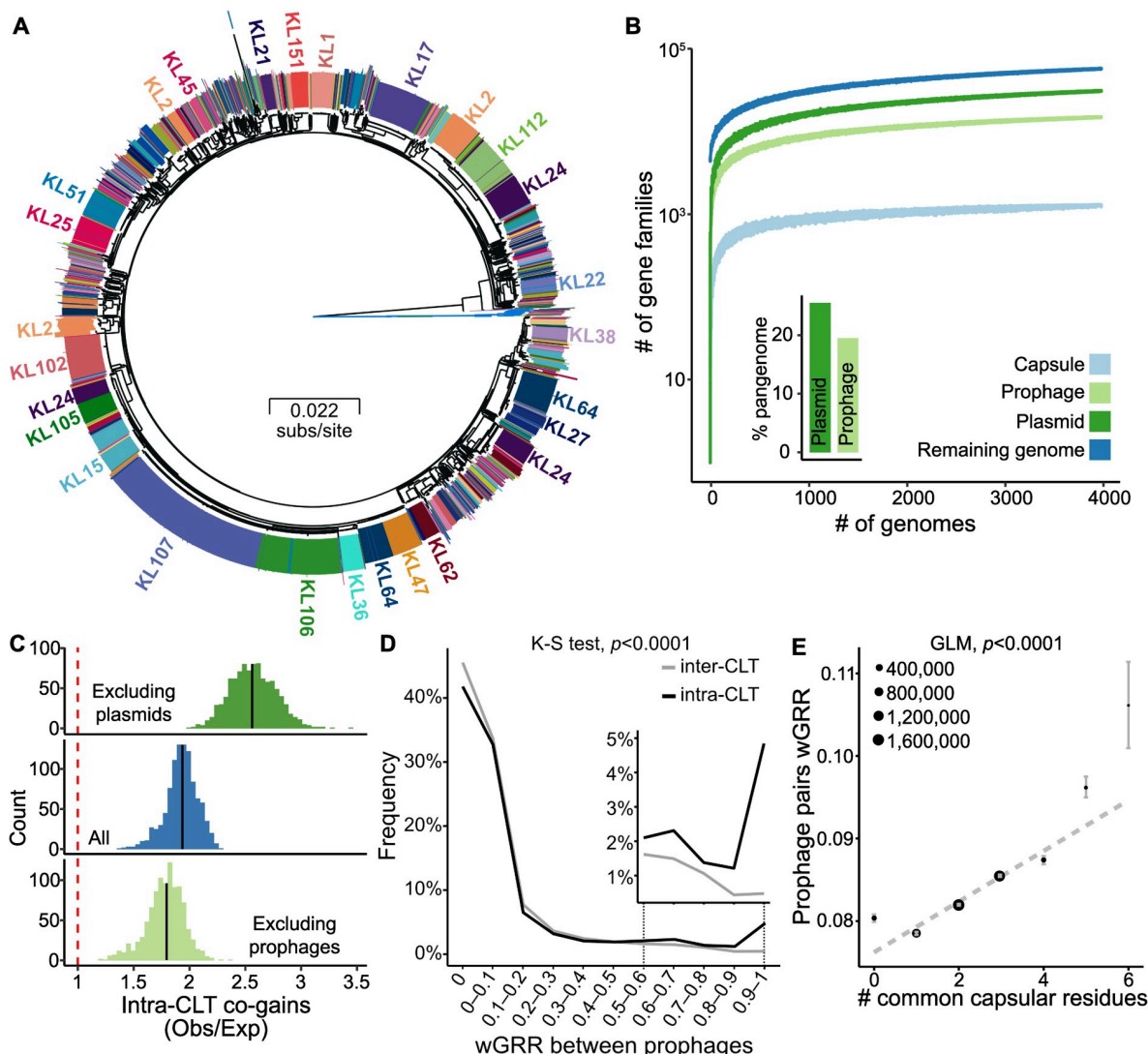

**Fig 1. Gene flow is higher between strains of the same serotype. (A)** Core genome phylogenetic tree with the 22 *Klebsiella quasipneumoniae* subsp. *similipneumoniae* (misannotated as Kpn in RefSeq) strains as an outgroup (blue branches). The annotation circle represents the 108 CLTs predicted by Kaptive. The largest clusters of CLTs (>20 isolates) are annotated (full list in https://doi.org/10.6084/m9.figshare.14673156). **(B)** Rarefaction curves of the pangenome of prophages, plasmids, capsule genes, and all remaining genes (Genome). The points represent 50 random samples for each bin (bins increasing by 10 genomes). The inset bar plot represents the percentage of gene families of the Kpn pangenome including genes of plasmids or prophages (https://doi.org/10.6084/m9.figshare.14673141). **(C)** Histograms of the excess of intra-CLT co-gains in relation to those observed inter-CLT (Observed/Expected ratio obtained by 1,000 simulations). The analysis includes all genes (center), excludes prophages (bottom), or excludes plasmids (top) (https://doi.org/10.6084/m9.figshare.14673147). **(D)** Gene repertoire relatedness between independently acquired prophages in bacteria of different (inter-CLT, gray) or identical CLT (intra-CLT, black). The insert is a zoom of the distribution for the highest values of wGRR (https://doi.org/10.6084/m9.figshare.14673144). **(E)** Linear regression of the wGRR between pairs of prophages and the number of capsular residues in common between their hosts. The points represent the mean for each category, with their size corresponding to the number of pairs per category. Error bars represent the standard error of the mean. The regression was performed on the original raw data, but only the averages are represented for clarity (https://doi.org/10.6084/m9.figshare.14673165). CLT, capsular locus type; GLM, generalized linear model; Kpn, *Klebsiella pneumoniae*; wGRR, gene repertoire relatedness weighted by sequence identity.

infecting prophages are 1.7 times more similar when in bacteria with identical rather than different CLTs (Fig 1D; two-sample Kolmogorov–Smirnov test, $p < 0.0001$). To confirm that phage-mediated HGT is favored between strains of the same CLT, we repeated the analysis of gene co-gains after removing the prophages from the pangenome. As expected, the preference

toward same-CLT exchanges decreased from 1.95× to 1.73× (Fig 1C). This suggests that HGT tends to occur more frequently between strains of identical serotypes than between strains of different serotypes, a trend that is amplified by the transfer of temperate phages.

Most of the depolymerases that allow phages to overcome the capsule barrier act on specific disaccharides or trisaccharides, independently of the remaining monomers [30–32]. This raises the possibility that phage-mediated gene flow could be higher between strains whose capsules have common oligosaccharide residues. To test this hypothesis, we compiled the information on the 76 capsular chemical structures previously described [33]. The genomes with these CLTs, 59% of the total, show a weak but significant proportionality between pro-phage similarity, and the number of similar residues in their host capsules (Fig 1E), i.e., pro-phages, are more similar between bacteria with more biochemically similar capsules.

## Recombination swaps biochemically related capsules

To understand the genetic differences between serotypes and how these could facilitate swaps, we compared the gene repertoires of the different capsular loci (between *galF* and *ugd*, S1 Fig). As expected from previous studies [11,17], this analysis revealed a clear discontinuity between intra-CLT comparisons that had mostly homologous genes and the other comparisons, where many genes (average = 10) lacked homologs across serotypes (Wilcoxon test, $p < 0.0001$, Fig 2A). As a result, the capsule pangenome contains 325 gene families that are specific to a CLT (out of 547, see "Pan- and persistent genomes"). This implies that serotype swaps require the acquisition of multiple novel genes by horizontal transfer. To quantify and identify these CLT swaps, we inferred the ancestral CLT in the phylogenetic tree and found a rate of 0.282 swaps per branch (see "Serotype swaps identification"). We then identified 103 highly confident swaps, some of which occurred more than once (Fig 2B). We used the chemical characterization of the capsules described above to test if swaps were more likely between serotypes with more similar chemical composition. Indeed, swaps occurred between capsules with an average of 2.42 common sugars (mean Jaccard similarity 0.54), more than the average value across all other possible CLT pairs (1.98, mean Jaccard similarity 0.38, Wilcoxon test, $p < 0.0001$, S2A Fig). Interestingly, the wGRR of the swapped loci is only 3% higher than the rest of pairwise comparisons (S2B Fig). This suggests that successful swaps are poorly determined by the differences in gene repertoires. Instead, they are more frequent between capsules that have more similar chemical composition.

The existence of a single capsule locus in Kpn genomes suggests that swaps occur by homologous recombination at flanking conserved sequences [11]. We used Gubbins [34] to detect recombination events in the 25 strains with terminal branch serotype swaps and closely related completely assembled genomes (see "Detection of recombination tracts"). We found long recombination tracts encompassing the capsule locus in 24 of these 25 genomes, with a median length of 100.3 kb (Fig 2C). At least one border of the recombination tract was less than 3-kb away from the capsule locus in 11 cases (46%). Using sequence similarity to identify the origin of the transfer, we found that most recombination events occurred between distant strains and no specific clade (Fig 2D). We conclude that serotype swaps occur by recombination at the flanking genes with DNA from genetically distant isolates but chemically related capsules.

## Capsule inactivation follows specific paths, might be driven by phage predation, and spurs HGT

We sought to investigate whether the aforementioned swaps occur by an intermediary step where cells have inactivated capsule loci. To do so, we first established the frequency of inactivated capsular loci. We used the Kaptive software to detect missing genes, expected to be

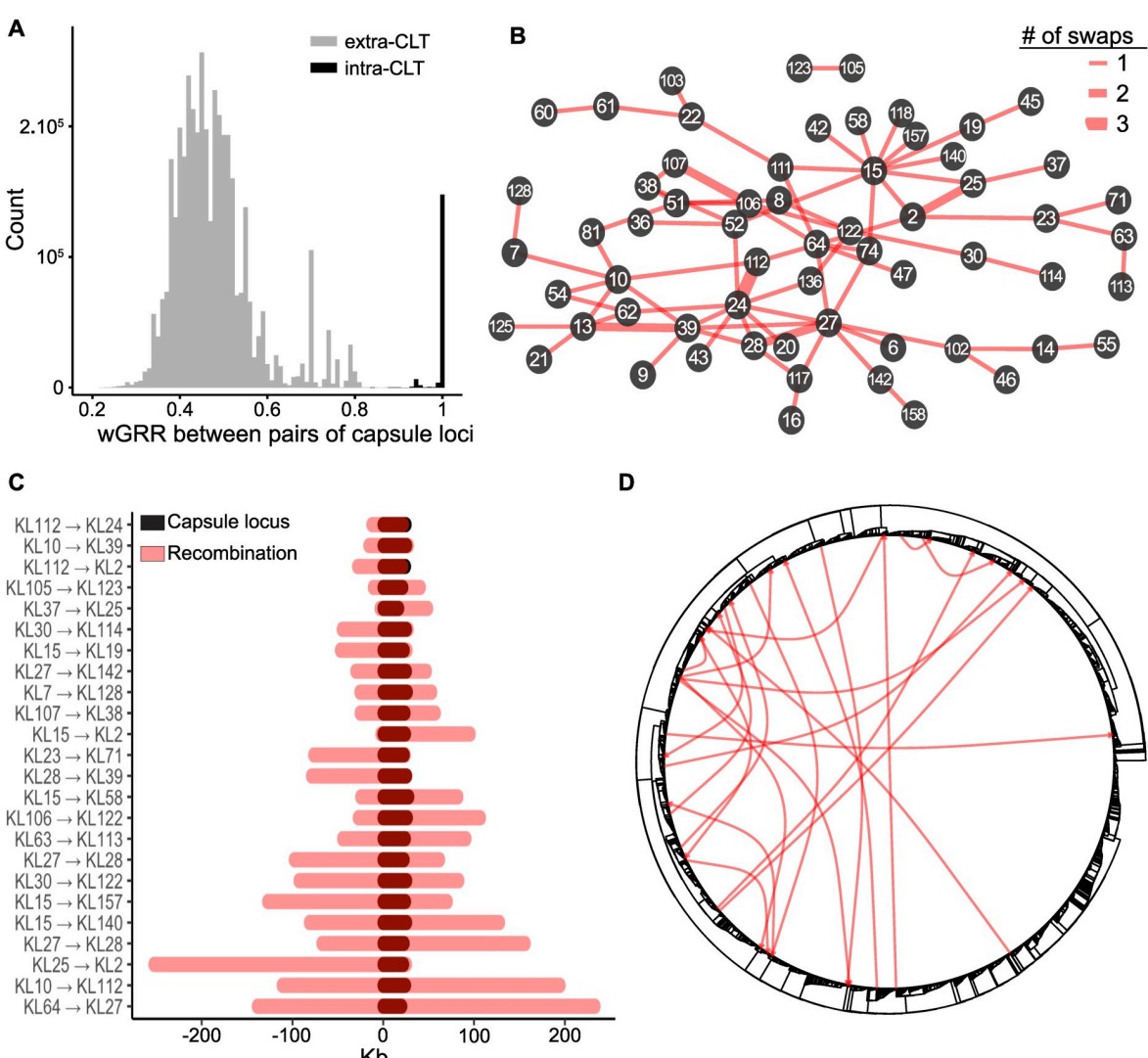

**Fig 2. Homologous recombination events lead to frequent CLT swaps. (A)** Histogram of the comparisons of gene repertoire relatedness (wGRR) between capsular loci of the same (intra-) or different (inter-) CLT (https://doi.org/10.6084/m9.figshare.14673171). **(B)** Network of CLT swaps identified by ancestral state reconstruction, with edge thickness corresponding to the number of swaps, and numbers **x** within nodes corresponding to the CLT (KL**x**). **(C)** Recombination encompassing the capsule locus detected with Gubbins. The positions of the tracts are represented in the same scale, where the first base of the *galF* gene was set as 0 (https://doi.org/10.6084/m9.figshare. 14673150). **(D)** Putative donor–recipient pairs involved in the CLT swaps of panel C indicated in the Kpn tree. CLT, capsular locus type; Kpn, *Klebsiella pneumoniae*; wGRR, gene repertoire relatedness weighted by sequence identity.

encoded in capsule loci found on a single contig. We also used the Kaptive database of capsular proteins to detect pseudogenes using protein–DNA alignments in all genomes. We found 55 missing genes and 447 pseudogenes, among 9% of the loci. The frequency of pseudogenes was not correlated with the quality of the genome assembly (see "Identification of capsule pseudogenes and inactive capsule loci"), and all genomes had at least a part of the capsule locus. We cannot exclude that some of these mutations fixated during passage prior to sequencing if these conditions strongly select for capsule loss. However, many isolates harbor several pseudogenes, which suggests that capsule inactivation is not due to very strong selection of one inactivating mutation during passage. We classed 11 protein families as essential for capsule production (Table A in S1 Text). At least one of these essential genes was missing in 3.5% of

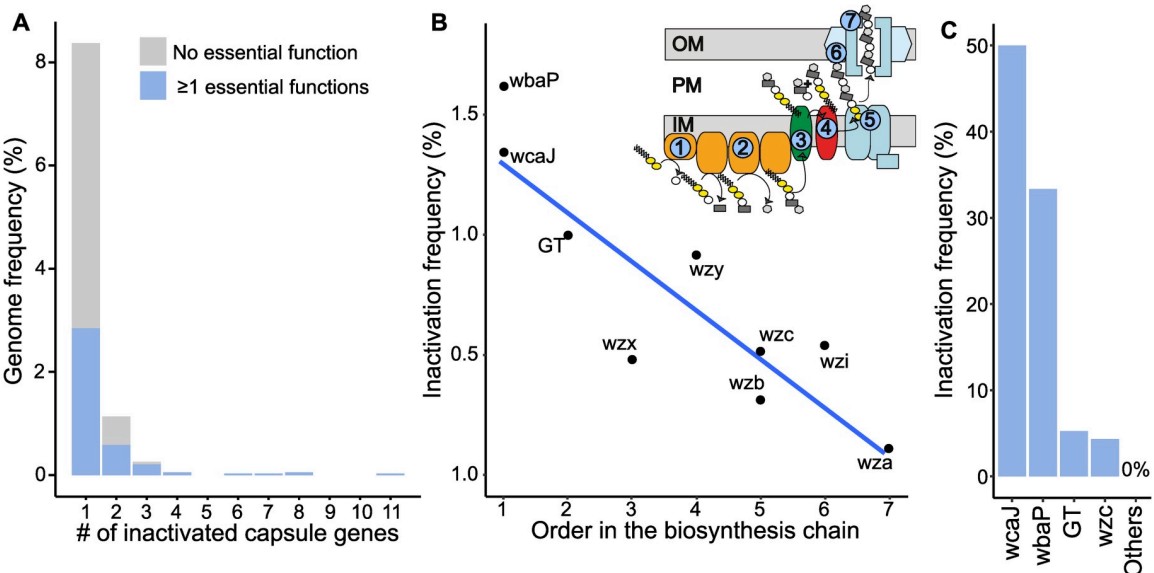

**Fig 3. Loss of function in the capsule locus.** (A) Distribution of the number of inactivated capsule genes per genome, split in 2 categories: loci lacking a functional essential capsule gene (blue, non-capsulated strains), and other loci only lacking nonessential capsule genes (white, not categorized as non-capsulated strains) (https://doi.org/10.6084/m9.figshare.14673153). (B) Linear regression between the inactivation frequency and the rank of each gene in the biosynthesis pathway ($p = 0.001$, $R^2 = 0.78$). The numbers in the scheme of the capsule assembly correspond to the order in the biosynthesis pathway (https://doi.org/10.6084/m9.figshare.14673153). (C) Frequency of inactivated capsule genes arising in the non-capsulated clones isolated in 8 different strains after approximately 20 generations in LB growth medium. Genomes containing several missing genes and pseudogenes are not included (https://doi.org/10.6084/m9.figshare.14673177). GT, glycosyl transferases; LB, Luria–Bertani.

the loci, which means these strains are likely non-capsulated (Fig 3A). These variants are scattered in the phylogenetic tree with no particular clade accounting for the majority of these variants (S3 Fig), e.g., there are non-capsulated strains in 61 of the 617 sequence types (STs) identified by Kleborate. These results suggest that capsule inactivation has little phylogenetic inertia, i.e., it is a trait that changes very quickly, either because the variants are counterselected or because capsules are quickly reacquired. Accordingly, we could not detect significant phylogenetic inertia with Pagel Lambda test [35] ($p > 0.05$). Hence, capsule inactivation is frequent, but non-capsulated lineages do not persist for long periods of time.

We further investigated the genetic pathways leading to capsule inactivation. Interestingly, we found that the pseudogenization frequency follows the order of biosynthesis of the capsule (linear regression, $p = 0.001$, $R^2 = 0.78$), with the first (*wbaP* or *wcaJ*) and second steps (glycosyl transferases, GT) being the most commonly inactivated when a single essential gene is a pseudogene (Fig 3B). The overall frequency of gene inactivation drops by 14% per rank in the biosynthesis chain. To confirm these results, we made several controls. To show that this is not simply an effect of gene length, we normalized the inactivation frequency by gene length and found the same relationship (S4A Fig, $p = 0.005$, $R^2 = 0.7$). To check that this was not a mutation hotspot induced by simple sequence repeats, which are prone to polymerase-slippage–induced mutations and sequencing errors, we searched for mono- and di-nucleotide tracts. We found that their frequency in the commonly inactivated pair *wcaJ*/*wbaP* was smaller than in the rarely mutated group of genes *wzc*/*wzx*/*wzy* (S4B Fig). Finally, we verified that our analysis was not impacted by a higher allelic diversity in these genes in the reference database. We found that the most genetically diverse genes were not the most inactivated (S4C Fig). Together, these results support the idea that selection plays a key role in the fixation of inactivating mutations in the early genes of the capsule biosynthetic pathway.

To test if similar results are found when capsules are counterselected in the laboratory, we analyzed a subset of populations stemming from a short evolution experiment in which populations of 8 different strains of *Klebsiella* spp. were diluted daily during 3 days (approximately 20 generations) under agitation in Luria–Bertani (LB), a medium known to select for capsule inactivation [36]. Each strain belongs to a different phylogenetic group (ST), encompassing 6 different serotypes and isolation sources (Table B in S1 Text). We have previously shown that under such conditions, phage pressure accelerates capsule loss [25]. After 3 days, non-capsulated clones emerged in 22 out of 24 populations from 8 different ancestral strains. We isolated one non-capsulated clone for Illumina sequencing from each population and searched for the inactivating mutations with the same pipeline as for the genomic dataset. We also compared our method with two popular tools used to detect new mutations in evolved isolates, *breseq* [37] and *snippy*, which rely on read mapping onto a reference genome. All 3 approaches yielded comparable conclusions, although some mutations were not found by all 3. Overall, 13 out of the 16 inactivated or deleted genes found by our method were also identified by either *breseq*, *snippy*, or both. Read mapping approaches detected other types of mutations, like intergenic mutations, and few mutations in the rest of the genome, which are not detectable by our targeted approach (https://doi.org/10.6084/m9.figshare.14673177). We found that most of these were localized in *wcaJ* and *wbaP* (Fig 3C). In accordance with our comparative genomics analysis, we found fewer loss-of-function mutations in GTs and *wzc* and none in the latter steps of the biosynthetic pathway, except for one large deletion event encompassing almost all the capsular locus (https://doi.org/10.6084/m9.figshare.14673177). Studies focusing on the mutations conferring phage resistance in Kpn have also reported an abundance of loss-of-function mutations in capsule genes leading to a non-capsulated phenotype [25,38], especially *wcaJ* [39,40]. These results strongly suggest that mutations leading to the loss of capsule production impose a fitness cost determined by the position of the inactivated gene in the biosynthesis pathway.

Once a capsule locus is inactivated, the function can be reacquired by (1) reversion mutations fixing the broken allele; (2) restoration of the inactivated function by acquisition of a gene from another bacterium, eventually leading to a chimeric locus; and (3) replacement of the entire locus leading to a CLT swap. Our analyses of pseudogenes provide some clues on the relevance of the 3 scenarios. We found 111 events involving nonsense point mutations. These could eventually be reversible (scenario 1) if the reversible mutation arises before other inactivating changes accumulate. We also observed 269 deletions (100 of more than 2 nucleotides) in the inactivated loci. These changes are usually irreversible in the absence of HGT. We then searched for chimeric loci (scenario 2), i.e., CLTs containing at least 1 gene from another CLT. We found 35 such loci, accounting for approximately 0.9% of the dataset (for example, a *wzc_KL1* allele in an otherwise KL2 loci), with only one occurrence of a *wcaJ* allele belonging to another CLT, and none for *wbaP*. Finally, the analysis of recombination tracts detailed above reveals frequent replacement of the entire locus between *galF* and *ugd* by recombination (S1 Fig, scenario 3).

Since reacquisition of the capsule function might often require HGT, we enquired if capsule inactivation was associated with higher rates of HGT. Indeed, the number of genes gained by HGT per branch of the phylogenetic tree is higher in branches where the capsule was inactivated than in the others (2-sample Wilcoxon test, $p < 0.0001$, Fig 4A), even if these branches have similar sizes (S6 Fig). We compared the number of phages and conjugative systems acquired in the branches where capsules were inactivated against the other branches. This revealed significantly more frequent (3.6 times more) acquisition of conjugative systems (Fisher exact test, $p < 0.0001$, Fig 4B) upon capsule inactivation. This was also the case, to a lesser extent, for prophages. Intriguingly, we observed even higher relative rates of acquisition

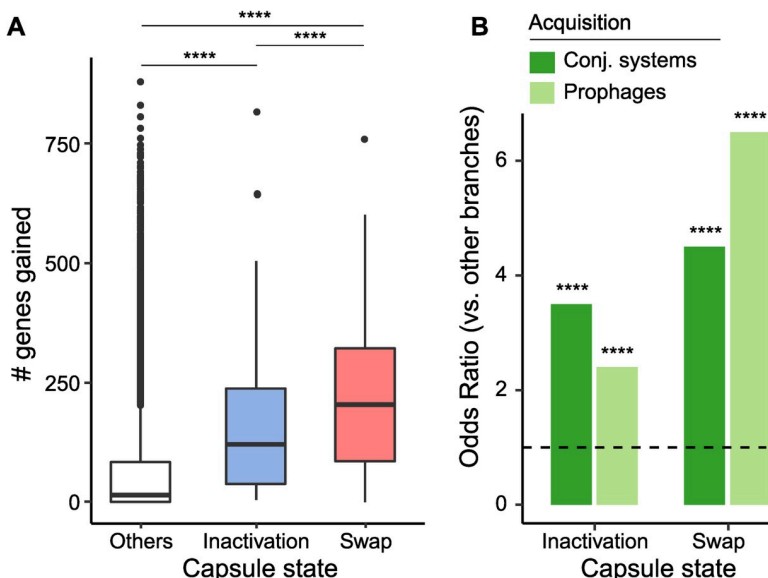

**Fig 4. Changes in capsule state impact HGT. (A)** Number of genes gained in terminal branches of the phylogenetic tree where the capsule was inactivated, swapped, and in the others (2-samples Wilcoxon tests). **(B)** Increase in the frequency of acquisition of prophages and conjugative systems on branches where the capsule was either inactivated or swapped, relative to the other branches, as represented by odds ratio (Fisher exact test). ****: *p*-value < 0.0001 (https://doi.org/10.6084/m9.figshare.14673159). HGT, horizontal gene transfer.

of genes and MGEs in branches where the serotype was swapped (Fig 4A and 4B). The latter branches are 2.7 times longer than the others (S6 Fig), which may be due to the recombination tracts associated with the swap. The difference in branch lengths between the swapped and inactivated categories are on the order of magnitude of the difference in the number of genes gained, suggesting approximately similar rates of gene gain in both categories. In contrast, the difference in terms of gained genes between branches with capsule swaps and branches where capsules remained unchanged is much larger than 2.7. Hence, branches where capsules were swapped, like those where they were inactivated, represent periods of more frequent acquisition of conjugative elements and phages relative to the periods where the capsule remained unchanged. The acquisition rate is particularly larger for conjugative systems in branches where the capsule was inactivated and larger for prophages in branches that swapped (6.5 versus 4.5 times more). Overall, the excess of HGT in periods of capsule inactivation facilitates the reacquisition of a capsule and the novel acquisition of other potentially adaptive traits.

## Conjugative systems are frequently transferred across serotypes

The large size of the Kpn capsule locus is difficult to accommodate in the phage genome, and the tendency of phages to be serotype specific makes them unlikely vectors of novel capsular loci. Also, the recombination tracts observed in Fig 2C are too large to be transduced by most temperate phages of Kpn, whose prophages average 46 kb [25]. Since Kpn is not naturally transformable, we hypothesized that conjugation is the major driver of capsule acquisition. Around 80% of the strains encode a conjugative system, and 94.4% have at least one. Plasmids alone make 25.5% of the pangenome (Fig 1B). To these numbers, one should add the genes present in integrative conjugative elements (ICEs). Unfortunately, there is currently no method to identify ICEs accurately in draft genomes. By subtracting the total number of conjugative elements from those identified in plasmids, we estimate that 41% of the conjugative systems in Kpn are not in plasmids but in ICEs. Since ICEs and conjugative plasmids have

approximately similar sizes [41], the joint contribution of ICEs and plasmids in the species pangenome is very large.

We identified independent events of infection by conjugative systems as we did for prophages (see above). The 5,144 conjugative systems fell into 252 families with 1,547 infection events. On average, pairs of conjugative systems acquired within the same CLT were only 3% more similar than those in different CLTs. This suggests that phage- and conjugation-driven HGT have very different patterns, since the former tend to be serotype specific, whereas the latter are very frequently transferred across serotypes. This opposition is consistent with the analysis of co-gains (Fig 1C), which were much more serotype dependent when plasmids were excluded from the analysis and less serotype dependent when prophages were excluded. To further test our hypothesis, we calculated the number of CLTs where one could find each family of conjugative systems or prophages and then compared these numbers with the expectation if they were distributed randomly across the species. The results show that phage families are present in much fewer CLTs than expected, whereas there is no bias for conjugative systems (Fig 5). We conclude that conjugation spreads plasmids across the species regardless of

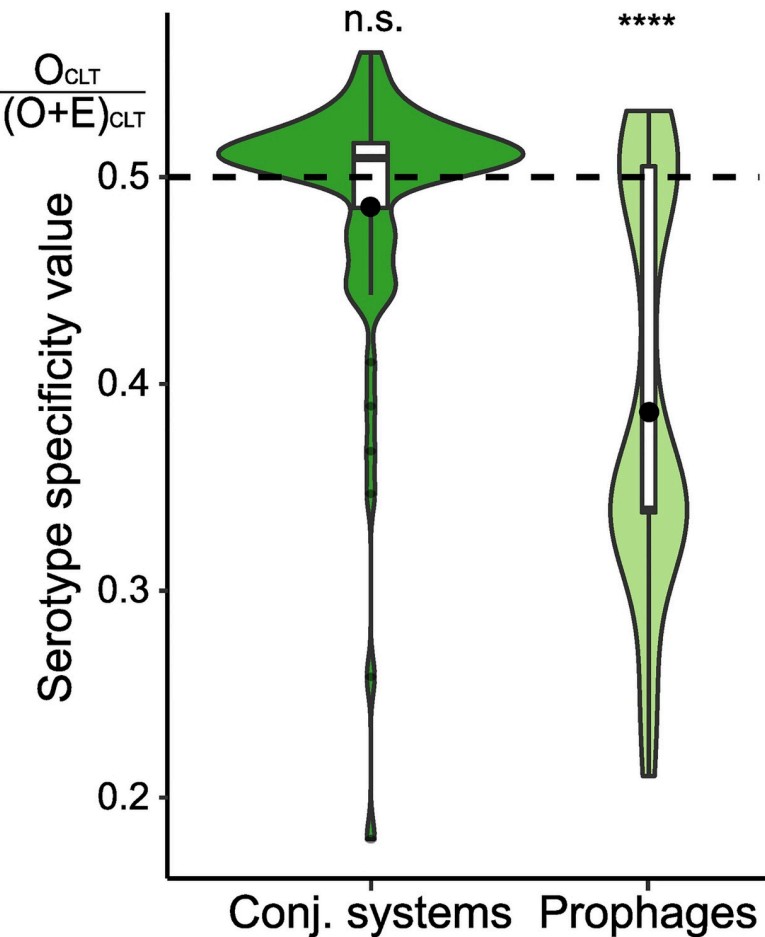

**Fig 5. Serotype specificity of prophages and conjugative systems.** $O_{CLT}$: observed number of serotypes infected per family of homologous element. $E_{CLT}$: expected number of serotypes infected per family of homologous element generated by 1,000 simulations (see "CLT specificity"). This measure aims to capture the number of different CLT that each prophage and conjugative system were able to infect, compared to what was expected by chance given the phylogeny. When the elements distribute randomly across CLTs, the value is 0.5 (dashed line). Very low values indicate high serotype specificity. One-sample Wilcoxon test. ****: $p$-value $< 0.0001$ (https://doi.org/10.6084/m9. figshare.14673162). CLT, capsular locus type.

serotype. Together, these results reinforce the hypothesis that conjugation drives genetic exchanges between strains of different serotypes, decreasing the overall bias toward same-serotype exchanges driven by phages.

## Capsule inactivation results in increased conjugation frequency

Together, these elements led us to hypothesize that capsule inactivation results in higher rates of acquisition of conjugative elements. This is consistent with the observation that terminal branches associated with inactive capsules have a higher influx of conjugative systems than prophages (Fig 4B). In the absence of published data on the impact of the capsule on conjugation frequency, we tested experimentally our hypothesis on a diverse set of *Klebsiella* isolates composed of four strains from different STs: three *Klebsiella pneumoniae sensu stricto* (serotypes K1 and K2) and one *Klebsiella variicola* (serotype K30, also found in Kpn). To test the role of the capsule in plasmid acquisition, we analyzed the conjugation frequency of these strains and their non-capsulated counterparts, deprived of *wcaJ*, the most frequent pseudogene in the locus both in the genome data and in our experimental evolution (see "Conjugation assay"). For this, we built a plasmid that is mobilized in *trans*, i.e., once acquired by the new host strain, it cannot further conjugate, due to their lack of a compatible conjugative system. This allows to measure precisely the frequency of conjugation between the donor and the recipient strain. In agreement with the results of the computational analysis, we found that the frequency of conjugation is systematically and significantly higher in the mutant than in the associated wild type (WT) for all four strains (paired Wilcoxon test, *p*-value = 0.002, Fig 6). On average, non-capsulated strains conjugated 8.06 times more than capsulated strains. In strain CG43, this difference was 20-fold.

Interestingly, the difference in conjugation rates between the mutants and their WT is inversely proportional to the conjugation frequency in the WT, possibly because some WT strains already conjugate at very high rates. Cell densities were normalized to the same optical density ($OD_{600} = 0.9$) before mating, and the donor culture was the same for all recipient strains per biological replicate. Cells were allowed to conjugate for one hour. Still, there were slightly fewer colony-forming unit (CFU; 3.2× less, paired Wilcoxon test, $p < 0.001$) on the membranes with non-capsulated mutants than on those with the WT. This may lead to a slight underestimation of the conjugation frequency in non-capsulated mutants. As a consequence, we may have underestimated the differences in conjugation frequency. Overall, these experiments show that the ability to receive a conjugative element is increased in the absence of a functional capsule. Hence, non-capsulated variants acquire more genes by conjugation than the others. Interestingly, if the capsule is transferred by conjugation, this implies that capsule inactivation favors the very mechanism leading to its subsequent reacquisition.

## Discussion

The specificity of many Kpn phages to one or a few chemically-related serotypes is presumably caused by their reliance on capsules to adsorb to the cell surface and results from the long-standing coevolution of phages with their Kpn hosts. One might invoke environmental effects to explain these results, since populations with identical or closely related serotypes might often co-occur and thus potentiate more frequent cross infections. However, the same ecological bias would be expected to increase the rates of conjugation between identical or closely related serotypes. This could not be detected, suggesting that bias in intra-serotype gene flow is mediated by phages. It is well known that some phages carry capsule depolymerases acting on disaccharides or trisaccharides [30–32] that may be similar across serotypes and thus favor transfer of phages between these cells. This fits our previous studies on the infection networks

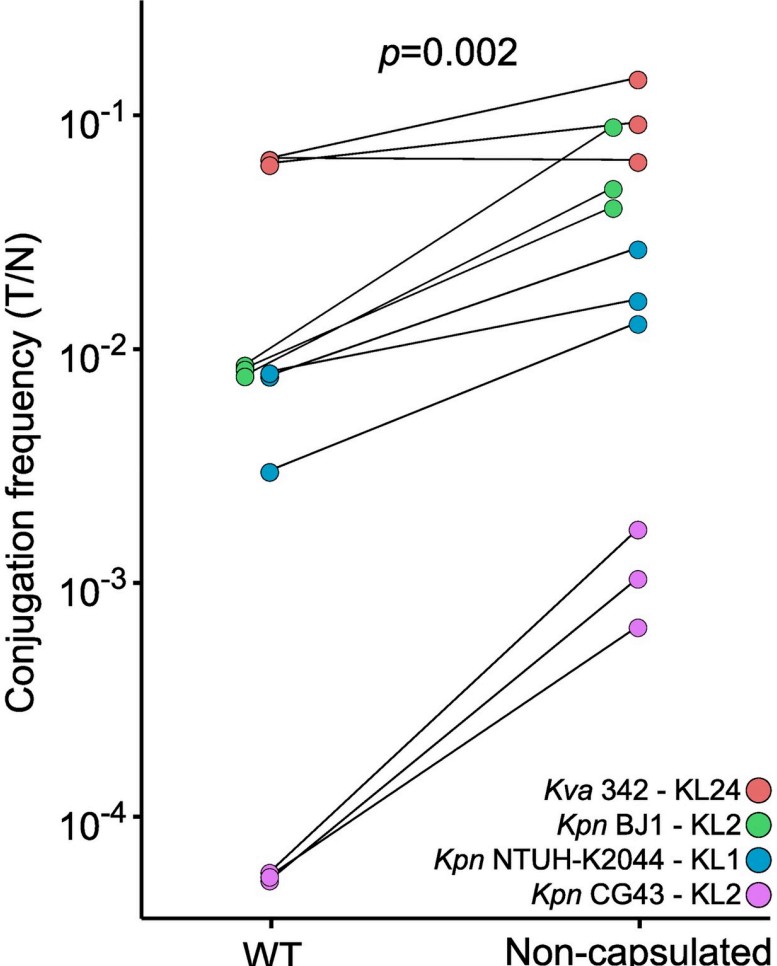

**Fig 6. Capsules negatively impact conjugation.** Conjugation frequency of WT and their associated non-capsulated (*ΔwcaJ*) mutants. The conjugation efficiency (T/N, see "Conjugation assay") is represented on a log scale. Each pair of points represents a biological replicate. Strains and their capsule locus type are listed in the order in which they appear from the top to the bottom of the plot. The *p*-value for the paired Wilcoxon test is displayed (https://doi.org/10.6084/m9.figshare.14673168). Kpn, *Klebsiella pneumoniae*; Kva, Klebsiella variicola; CG, clonal group; WT, wild type.

of Kpn prophages [25] and suggests that a population of cells encoding and expressing a given serotype has more frequent phage-mediated genetic exchanges with bacteria of identical or chemically similar serotypes (Fig 7A). In this context, phages carrying multiple capsule depolymerases have broader host range and may have a key role in phage-mediated gene flow between very distinct serotypes. For example, one broad host range virulent phage has been found to infect 10 distinct serotypes because it encodes an array of at least 9 depolymerases [24].

The interplay between the capsule and conjugative elements has been much less studied. Our comparative genomics analyses reveal that conjugation occurs across strains of identical or different CLTs at similar rates. Furthermore, our experimental data shows that non-capsulated mutants are up to 20 times more receptive to plasmid conjugation than the capsulated WT bacteria, an effect that seems more important for the WT that are poor recipients. These results are likely to be relevant not only for non-capsulated strains, but also for those not expressing the capsule at a given moment. If so, repression of the expression of the capsule may allow bacteria to escape phages and endure extensive acquisition of conjugative elements.

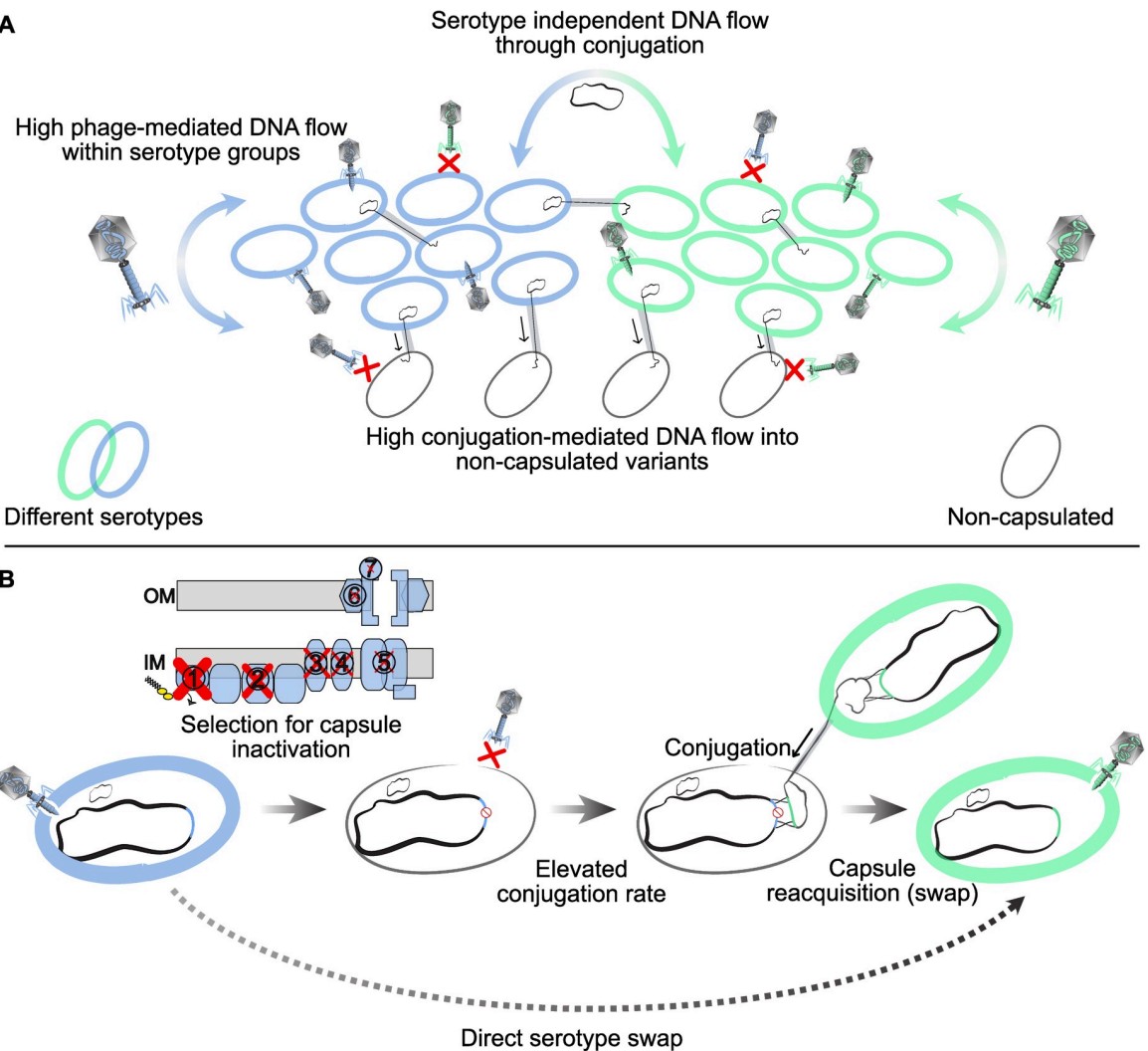

**Fig 7. A model for the interplay between serotypes and mobile elements. (A)** The capsule impacts Kpn gene flow. A bacterial population expressing a given serotype (blue or green) preferentially exchanges DNA by phage-mediated processes with bacteria of identical or chemically similar serotypes. Such flow may be rare toward non-capsulated bacteria because they are often resistant to Kpn phages. In contrast, conjugation occurs across serotypes and is more frequent to non-capsulated bacteria. **(B)** Proposed model for serotype swaps in Kpn. The capsule locus is colored according to its type. Capsule inactivation is occasionally adaptive. The pseudogenization process usually starts by the inactivation of the genes involved in the early stages of the capsule biosynthesis, as represented by the size of the red cross on the capsule assembly scheme. Non-capsulated strains are often protected from Kpn phage infections while acquiring more genes by conjugation. This increases the likelihood of capsule reacquisition. Such reacquisition can bring a new serotype, often one that is chemically similar to the previous one, and might be driven by conjugation because of its high frequency in non-capsulated strains. Serotype swaps rewire phage-mediated genetic transfers. IM, inner membrane; Kpn, *Klebsiella pneumoniae*; OM, outer membrane.

These results may also explain a long-standing conundrum in Kpn. The hypervirulent lineages of Kpn, which are almost exclusively of serotypes K1 and K2 [10,42], have reduced pangenome, plasmid, and capsule diversity. They also often carry additional factors like *rmpA* upregulating the expression of the capsule [42]. In contrast, they are very rarely multidrug resistant. Our data suggest that the protection provided by thick capsules hampers the acquisition of conjugative elements, which are the most frequent vectors of antibiotic resistance. Furthermore, the moments of capsule swap, repression of expression, or inactivation are expected to be particularly deleterious for hypervirulent clones, because the capsule is a virulence factor,

thus further hampering their ability to acquire the conjugative MGEs that carry antibiotic resistance. This may have favored a specialization of the clones into either hypervirulent or multidrug resistance. Unfortunately, the capsule is not an insurmountable barrier for conjugation, and recent reports have uncovered the emergence of multidrug-resistant hypervirulent clones [43,44]. The most worrisome pattern of evolution of such strains is the fusion of the resident virulence plasmids of hypervirulent clones with highly mobile multidrug resistance plasmids [45]. Strikingly, the recently isolated MS3802 clone, which belongs to the hypervirulent lineage ST23 and harbors a chimeric virulence/resistance plasmid, is string test negative and encodes a strongly degraded KL1 capsule locus [46]. Capsule inactivation may have facilitated the acquisition of a conjugative resistance plasmid that co-integrated with the resident virulence plasmid.

We observed that branches of isolates lacking a functional capsule have higher rates of acquisition of conjugative systems than prophages, whereas those where there was a capsule swap have the inverse pattern (Fig 4B). What could justify these differences in the interplay of the capsule with phages and conjugative elements? Phages must adsorb on the cell surface, whereas there are no critical positive determinants for incoming conjugation pilus [47]. As a result, serotype swaps may affect much more the flow of phages than that of conjugative elements. Capsule inactivation may have an opposite effect on phages and plasmids: it removes a point of cell attachment for phages, decreasing their infection rates, and removes a barrier to the conjugative pilus, increasing their ability to transfer DNA. Hence, when a bacterium has an inactivated capsule, e.g., because of phage predation, it becomes more permissive for conjugation. In contrast, when a bacterium acquires a novel serotype, it may become sensitive to novel phages resulting in rapid turnover of its prophage repertoire. These results implicate that conjugation should be much more efficient at spreading traits across the entire Kpn species than phage-mediated mechanisms, whose role would be important for HGT between strains of similar or closely related serotypes (Fig 7A).

The existence of serotype swaps has been extensively described in the literature for Kpn [16] and many other species [48,49]. Whether these swaps imply a direct serotype replacement, or an intermediate non-capsulated state, is not sufficiently known. Several processes are known to select for capsule inactivation in some bacteria, including growth in rich medium [36], phage pressure, and immune response [25,26,39,50] (Fig 7B). Because of the physiological effects of these inactivations, and their impact on the rates and types of HGT, it is important to quantify the frequency of inactivated (or silent) capsular loci and the mechanisms favoring it. Our study of pseudogenization of capsular genes revealed a few percent of putative non-capsulated strains scattered in the species tree, opening the possibility that non-capsulated strains are a frequent intermediate step of serotype swap. The process of capsule inactivation is shaped by the capsule biosynthesis pathway, the frequency of pseudogenization decreasing linearly with the rank of the gene in the capsule biosynthesis pathway. This suggests a major role for epistasis in the evolutionary pathway leading to non-capsulated strains. Notably, the early inactivation of later genes in the biosynthesis pathway, while the initial steps are still functional, can lead to the sequestration of key molecules at the cell envelope or the toxic accumulation of capsule intermediates (Fig 7B). Accordingly, Δ*wza* and Δ*wzy* mutants, but not Δ*wcaJ*, lead to defects in the cell envelope of the strain Kpn SGH10 [51]. In *Acinetobacter baumannii*, high-density transposon mutagenesis also recently revealed that inactivation of genes involved in the last steps of capsule biosynthesis is much more deleterious than those encoding the early steps [52].

Capsule reacquisition is more likely driven by conjugation than by phages, which are generally dependent on the capsule to infect their host. Hence, the increased rate of acquisition of conjugative elements by non-capsulated strains may favor the process of capsule reacquisition,

and, eventually, serotype swap. Cycles of gain and loss of capsular loci have been previously hypothesized in the naturally transformable species *Streptococcus pneumoniae*, because vaccination leads to counterselection of capsulated strains, and natural transformation seems to increase recombination in non-capsulated clades [53]. Accordingly, tracts with a median length 42.7 kb encompassing the capsule locus were found in serotype-swapped *S. pneumoniae* isolates [54]. In Kpn, such tracts were more than twice larger, which is consistent with the role of conjugation in capsule swap. Interestingly, these swaps are more frequent between CLTs encoding serotypes with common sugar residues, independently of their overall genetic relatedness. Understanding this result will require further work, but it suggests that genomic adaptation to the production of specific activated sugars can lead to genetic incompatibilities with other capsular genes.

Our results are relevant to understand the interplay between the capsule and other mobile elements in Kpn or other bacteria. We expect to observe more efficient conjugation when the recipient bacteria lack a capsule in other species. For example, higher conjugation rates of non-capsulated strains may help explain their higher recombination rates in *Streptococcus* [55]. The serotype specificity of phages also opens intriguing possibilities for them and for virion-derived elements. For instance, some *Escherichia coli* strains able to thrive in freshwater reservoirs have capsular loci acquired in a single-block horizontal transfer from Kpn [56]. This could facilitate interspecies phage infections (and phage-mediated HGT), since Kpn phages may now be adsorbed by these strains. Gene transfer agents (GTA) are co-options of virions for intraspecies HGT that are frequent among alpha-proteobacteria [57] (but not yet described in Kpn). They are likely to have equivalent serotype specificity, since they attach to the cell envelope using structures derived from phage tails. Indeed, the infection by the *Rhodobacter capsulatus* GTA model system depends on the host Wzy capsule [58], and non-capsulated variants of this species are phage resistant [59] and impaired in GTA-mediated transfer [60]. Our general prediction is that species where cells tend to be capsulated are going to coevolve with phages, or phage-derived tail structures, such that the latter will tend to become serotype specific. We speculate that future developments on the systematic detection of depolymerase genes will shed light on depolymerase swaps between phages, as a reciprocal phenomenon to capsule serotype swaps. Here, we could not make such an analysis since we identified very few depolymerases. Similar difficulties to find depolymerases were recently observed [25,61], suggesting that many such proteins remain to be identified.

These predictions have an impact in the evolution of virulence and antimicrobial therapy. Some alternatives to antibiotics—phage therapy, depolymerases associated with antibiotics, pyocins, and capsular polysaccharide vaccines—may select for the inactivation of the capsule [25,38,39]. Such non-capsulated variants have often been associated with better disease outcomes [62], lower antibiotic tolerance [21], and reduced virulence [20]. However, they can also be more successful colonizers of the urinary tract [63]. Our results suggest that these non-capsulated variants are at higher risk of acquiring resistance and virulence factors through conjugation, because ARGs and virulence factors are often found in conjugative elements in Kpn and in other nosocomial pathogens. Conjugation may also eventually lead to the reacquisition of functional capsules. At the end of the inactivation–reacquisition process, recapitulated on Fig 7, the strains may be capsulated, more virulent, and more antibiotic resistant.

## Materials and methods

### Genomes

We used the PanACoTa tool to generate the dataset of genomes [64]. We downloaded all the 5,805 genome assemblies labeled as *Klebsiella pneumoniae sensu stricto* (*Kpn*) from NCBI

RefSeq (accessed on October 10, 2018). We removed lower-quality assemblies by excluding genomes with L90>100. The pairwise genetic distances between all remaining genomes of the species was calculated by order of assembly quality (L90) using MASH [65]. Strains that were too divergent (MASH distance >6%) to the reference strain or too similar (<0.0001) to other strains were removed from further analysis. The latter tend to have identical capsule serotypes, and their exclusion does not eliminate serotype swap events. This resulted in a dataset of 3,980 strains which were re-annotated with *prokka* (v1.13.3) [66] to use consistent annotations in all genomes. Erroneous species annotations in the GenBank files were corrected using Kleborate (https://github.com/katholt/Kleborate). This step identified 22 *Klebsiella quasipneumoniae* subspecies *similipneumoniae* (Kqs) genomes that were used to root the species tree and excluded from further analyses. The accession number for each analyzed genome is presented in https://doi.org/10.6084/m9.figshare.14673156, along with all the annotations identified in this study.

## Pan- and persistent genome

The pangenome is the full repertoire of homologous gene families in a species. We inferred the pangenome with the connected-component clustering algorithm of MMSeqs2 (*release 5*) [67] with pairwise bidirectional coverage >0.8 and sequence identity >0.8. The persistent genome was built from the pangenome, with a persistence threshold of 99%, meaning that a gene family must be present in single copy in at least 3,940 genomes to be considered persistent. Among the 82,730 gene families of the Kpn pangenome, there were 1,431 gene families present in 99% of the genomes, including the Kqs. We used mlplasmids to identify the "plasmid" contigs (default parameters, species "*Klebsiella pneumoniae*" [68]). To identify the pangenome of capsular loci present in the Kaptive database, we used the same method as above, but we lowered the sequence identity threshold to >0.4 to put together more remote homologs.

## Phylogenetic tree

To compute the species phylogenetic tree, we aligned each of the 1,431 protein families of the persistent genome individually with mafft (v7.407) [69] using the option *FFT-NS-2*, back-translated the sequences to DNA (i.e., replaced the amino acids by their original codons) and concatenated the resulting alignments. We then made the phylogenetic inference using *IQ-TREE* (*v1.6.7.2*) [70] using ModelFinder (-m TEST) [71] and assessed the robustness of the phylogenetic inference by calculating 1,000 ultra-fast bootstraps (-bb 1,000) [72]. There were 220,912 parsimony-informative sites over a total alignment of 1,455,396 bp, and the best-fit model without gamma correction was a general time-reversible model with empirical base frequencies allowing for invariable sites (GTR+F+I). We did not use the gamma correction because of branch length scaling issues, which were 10 times longer than with simpler models, and is related to an optimization problem with big datasets in IQ-TREE. The tree is very well supported, since the average ultra-fast bootstrap support value was 97.6% and the median was 100%. We placed the Kpn species root according to the outgroup formed by the 22 *Kqs* strains. The tree, along with Kleborate annotations, can be visualized and manipulated in https://microreact.org/project/kk6mmVEDfa1o3pGQSCobdH/9f09a4c3.

## Capsule locus typing

We used Kaptive [17] with default options and the "K locus primary reference" to identify the CLT of strains. The predicted CLT is assigned a confidence level, which relies on the overall alignment to the reference CLT, the allelic composition of the locus, and its fragmentation level. We assigned the CLT to "unknown" when the confidence level of Kaptive was indicated

as "none" or "low," as suggested by the authors of the software. This only represented 7.9% of the genomes. After this filtering, we simply considered that 2 CLT are the same if they are both annotated with the same KLx name.

## Identification of capsule pseudogenes and inactive capsule loci

We first compiled the list of missing expected genes from Kaptive, which is only computed by Kaptive for capsule loci encoded in a single contig. Then, we used the Kaptive reference database of Kpn capsule loci to retrieve capsule reference genes for all the identified serotypes. We searched for sequence similarity between the proteins of the reference dataset and the 3,980 genome assemblies using blastp and tblastn (v.2.9.0) [73]. We then searched for the following indications of pseudogenization: stop codons resulting in protein truncation, frameshift mutations, insertions, and deletions (https://doi.org/10.6084/m9.figshare.14673153). Truncated and frameshifted coding sequences covering at least 80% of the original protein in the same reading frame were considered functional. Additionally, a pseudogene did not result in a classification of inactivated function if we could identify an intact homolog or analog. For example, if *wcaJ_KL1* has a frameshift, but *wcaJ_KL2* was found in the genome, the pseudogene was flagged and not used to define non-capsulated mutants. Complete gene deletions were identified by Kaptive among capsular loci encoded on a single contig. We built a dictionary of genes that are essential for capsule production by gathering a list of genes (annotated as the gene name in the Kaptive database) present across all CLTs and which are essential for capsule production according to experimental evidence (Table A in S1 Text). The absence of a functional copy of these essential genes resulted in the classification "non-capsulated" (except *wcaJ* and *wbaP*, which are mutually exclusive). To correlate the pseudogenization frequency with the order in the capsular biosynthesis process, we first sought to identify all glycosyl transferases from the different CLTs and grouped them in one category. To do so, we retrieved the GO molecular functions listed on UniProtKB of the genes within the Kaptive reference database. For the genes that could be ordered in the biosynthesis chain (Table A in S1 Text), we computed their frequency of inactivation by dividing the count of inactivated genes by the total number of times it is present in the dataset.

To test that sequencing errors and contig breaks were not leading to an excess of pseudogenes in certain genomes, we correlated the number of pseudogenes (up to 11) and missing genes with 2 indexes of sequence quality, namely, the sequence length of the shortest contig at 50% of the total genome length (N50) and the smallest number of contigs whose length sum makes up 90% of genome size (L90). We found no significant correlation in both cases (Spearman correlation test, $p$-values $> 0.05$), suggesting that our results are not strongly affected by sequencing artifacts and assembly fragmentation.

## Genetic similarity

We searched for sequence similarity between all proteins of all prophages or conjugative systems using MMSeqs2 with the sensitivity parameter set to 7.5. The hits were filtered (e-value $< 10^{-5}$, $\geq 35\%$ identity, coverage $>50\%$ of the proteins) and used to compute the set of bidirectional best hits (BBH) between each genome pair. BBH were used to compute the gene repertoire relatedness between pairs of genomes (weighted by sequence identity):

$$\text{wGRR}_{A,B} = \sum_i \frac{id(A_i, B_i)}{\min(\#A, \#B)},$$

as previously described [74], where $A_i$ and $B_i$ are the pair $i$ of homologous proteins present in A and B (containing respectively $\#A$ and $\#B$ proteins), $id(A_i, B_i)$ is their sequence identity, and

min(#*A*, #*B*) is the number of proteins of the element encoding the fewest proteins (#*A* or #*B*). wGRR varies between 0 and 1. It amounts to 0 if there are no homologous proteins between the genomes, and one if all genes of the smaller genome have a homolog in the other genome. Hence, the wGRR accounts for both frequency of homology and degree of similarity among homologs.

### Inference of genes ancestral states

We inferred the ancestral state of each pangenome family with PastML (v1.9.23) [75] using the maximum-likelihood algorithm MPPA and the F81 model. We also tried to run Count [76] with the ML method to infer gene gains and losses from the pangenome, but this took a prohibitive amount of computing time. To check that PastML was producing reliable results, we split our species tree (*cuttree* function in R, package stats) in 50 groups and for the groups that took less than a month of computing time with Count (2,500 genomes), we compared the results of Count to those of PastML. The 2 methods were highly correlated in term of number of inferred gains per branch (Spearman correlation test, Rho = 0.88, *p*-value < 0.0001). We used the results of PastML, since it was much faster and could handle the whole tree in a single analysis. Since the MPPA algorithm can keep several ancestral states per node if they have similar and high probabilities, we only counted gene gains when both ancestral and descendant nodes had one single distinct state (absent ➜ present).

### Analysis of conjugative systems

To detect conjugative systems, Type IV secretion systems, relaxases, and infer their MPF types, we used TXSScan with default options [77]. We then extracted the protein sequence of the conjugation systems and used these sequences to build clusters of systems by sequence similarity. We computed the wGRR (see "Genetic similarity") between all pairs of systems and clustered them in wGRR families by transitivity when the wGRR was higher than 0.99. This means that some members of the same family can have a wGRR <0.99. This threshold was defined based on the analysis of the shape of the distribution of the wGRR (S5A Fig). We used a reconstruction of the presence of members of each gene family in the species phylogenetic tree to infer the history of acquisition of conjugative elements (see "Inference of genes ancestral state"). To account for the presence of orthologous families, i.e., those coming from the same acquisition event, we kept only 1 member of a wGRR family per acquisition event. For example, if a conjugative system of the same family is present in 4 strains, but there were 2 acquisition events, we randomly picked 1 representative system for each acquisition event (in this case, 2 elements, 1 per event). Elements that resulted from the same ancestral acquisition event are referred as orthologous systems. We combined the predictions of mlplasmids and TXSScan to separate conjugative plasmids from ICEs. The distribution of conjugative system's MPF type in the chromosomes and plasmids is shown in S7 Fig.

### Prophage detection

We used PHASTER [78] to identify prophages in the genomes (accessed in December 2018). The category of the prophage is given by a confidence score that corresponds to "intact," "questionable," or "incomplete." We kept only the "intact" prophages because other categories often lack essential phage functions. We further removed prophage sequences containing more than 3 transposases after annotation with ISFinder [79] because we noticed that some loci predicted by PHASTER were composed of arrays of insertion sequences. We built clusters of nearly identical prophages with the same method used for conjugative systems. The wGRR threshold for clustering was also defined using the shape of the distribution (S5B Fig). The

definition of orthologous prophages follows the same principle than that of conjugative systems, they are elements that are predicted to result from one single past event of infection.

## Serotype swaps identification

We inferred the ancestral state of the capsular CLT with PastML using the maximum-likelihood algorithm MPPA, with the recommended F81 model [75]. In the reconstruction procedure, the low confidence CLTs were treated as missing data. This analysis revealed that serotype swaps happen at a rate of 0.282 swaps per branch, which are, on average, 0.000218 substitutions/site long in our tree. CLT swaps were defined as the branches where the descendant node state was not present in the ancestral node state. In 92% of the swaps identified by MPPA, there was only 1 state predicted for both ancestor and descendant node, and we could thus precisely identify the CLT swaps. These swaps were used to generate the network in Fig 3A.

## Detection of recombination tracts

We detected recombination tracts with Gubbins v2.4.0 [34]. Our dataset is too large to build one meaningful whole-genome alignment (WGA). Gubbins is designed to work with closely related strains, so we split the dataset into smaller groups defined by a single ST. We then aligned the genomes of each ST with snippy v4.3.8 (https://github.com/tseemann/snippy), as recommended by the authors in the documentation. The reference genome was picked randomly among the complete assemblies of each ST. We analyzed the 25 groups in which a CLT swap happened (see above) and for which a complete genome was available as a reference. We launched Gubbins independently for each WGA, using default parameters. We focused on the terminal branches to identify the recombination tracts resulting in CLT swap. We enquired on the origin of the recombined DNA using a sequence similarity approach. We used blastn [73] (-task megablast) to find the closest match of each recombination tract by querying the full tract against our dataset of genome assemblies and mapped the closest match based on the bit score onto the species tree.

## Identification of co-gains

We used the ancestral state reconstruction of the pangenome families to infer gene acquisitions at the terminal branches. We then quantified how many times an acquisition of the same gene family of the pangenome (i.e., co-gains) occurred independently in genomes of the same CLT. This number was compared to the expected number given by a null model where the CLT does not impact the gene flow. The distribution of the expectation of the null model was made by simulation in R, taking into account the phylogeny and the distribution of CLTs. In each simulation, we used the species tree to randomly redistribute the CLT trait on the terminal branches (keeping the frequencies of CLTs equal to those of the original data). We ran 1,000 simulations and compared them with the observed values with a 1-sample Z-test [80]:

$$Acquisition\ specificity\ score = \sum_g \frac{I_g \times (I_g - 1)}{T_g \times (T_g - 1)},$$

where the numerator is the number of pairs with gains in a CLT, and the denominator is the number of all possible pairs. With each gene family of the pangenome $g$, the number of gene gains in strains of the same CLT $I$, and the total number of gene gains $T$. This corresponds to the sum of total number of co-gains within a CLT, normalized by the total number of co-gains for each gene. This score captures the amount of gene acquisitions that happened within

strains of the same CLT. If the observed score is significantly different than the simulations assuming random distribution, it means that there was more genetic exchange within CLT groups than expected by chance.

## CLT specificity

We used the ancestral reconstruction of the acquisition of prophages and conjugative systems to count the number of distinct CLT in which such an acquisition occurred. For example, one prophage family can be composed of 10 members, coming from 5 distinct infection events in the tree: 2 in KL1 bacteria and 3 in KL2 bacteria. Therefore, we count 5 acquisitions in 2 CLT ($CLT_{obs}$ = 2). The null model is that of no CLT specificity. The distribution of the expected number of CLT infected following the null model was generated by simulation ($n$ = 1,000), as described above (see "Identification of co-gains"), and we plotted the specificity score as follows:

$$\text{Specificity score} = \frac{CLT_{obs}}{(CLT_{obs} + \overline{CLT_{exp}})},$$

where $CLT_{obs}$ is the observed number of CLT infected, and $\overline{CLT_{exp}}$ is the mean number of CLT infected in the simulations. Thus, the expected value under nonspecificity is 0.5.

Under our example of 5 acquisitions in 2 CLT ($CLT_{obs}$ = 2), a $\overline{CLT_{exp}}$ of 2 would mean that there is no difference between the observed and expected distributions across CLTs, and the specificity score is 0.5. Values lower than 0.5 indicate a bias toward regrouping of elements in a smaller than expected number of CLTs, whereas values higher than 0.5 indicate over-dispersion across CLTs. The statistics computed on Fig 5 are the comparison of all the specificity scores for all the prophage and conjugative systems families to the null model (score = 0.5) with a 1-sample Wilcoxon signed rank test.

## Handling of draft assemblies

Since more than 90% of our genome dataset is composed of draft assemblies, i.e., genomes composed of several chromosomal contigs, we detail here the steps undertaken to reduce the impact of such fragmentation on our analysis. We only included prophages and conjugative systems that are localized on the same contig (see "Prophage detection" and "Analysis of conjugative systems"). Kaptive is able to handle draft assemblies and adjust the confidence score accordingly when the capsule locus is fragmented, so we relied on the Kaptive confidence score to annotate the CLT, which was treated as missing data in all the analysis when the score was below "Good" (see "Capsule locus typing"). For the detection of missing capsular genes, performed by Kaptive, we verified that only non-fragmented capsular loci are included (see "Identification of capsule pseudogenes and inactive capsule loci"). For the detection of capsule pseudogenes, we included all assemblies and flagged pseudogenes that were localized on the border of a contig (last gene on the contig). Out of the 502 inactivated/missing genes, 47 were localized at the border of a contig. We repeated the analysis presented on Fig 3B after removing these pseudogenes and found an even better fit for the linear model at $R^2$ = 0.77 and $p$ = 0.004. Of note, such contig breaks are likely due to IS insertions, forming repeated regions that are hard to assemble, so we kept them in the main analysis.

## Analyses of lab-evolved non-capsulated clones

To pinpoint the mechanisms by which a diverse set of strains became non-capsulated, we took advantage of an experiment performed in our lab and described previously in [36]. Briefly, 3

independent overnight cultures of 8 strains (Table B in S1 Text) were diluted 1:100 into 5 mL of fresh LB and incubated at 37˚C under agitation. Each independent population was diluted 1:100 into fresh LB every 24 hours for 3 days (approximately 20 generations). We then plated serial dilutions of each population. A single non-capsulated clone per replicate population was isolated based on their translucent colony morphology, except in 2 replicate populations where all colonies plated were capsulated. We performed DNA extraction with the guanidium thiocyanate method [81], with modifications. DNA was extracted from pelleted cells grown overnight in LB supplemented with 0.7 mM EDTA. Additionally, RNAse A treatment (37˚C, 30 minutes) was performed before DNA precipitation. Each clone ($n = 22$) was sequenced by Illumina with 150pb paired-end reads, yielding approximately 1 GB of data per clone. The reads were assembled with Unicycler v0.4.4 [82], and the assemblies were checked for pseudogenes (see "Identification of capsule pseudogenes and inactive capsule loci"). We also used *breseq* [37] and *snippy* (https://github.com/tseemann/snippy) to verify that there were no further undetected mutations in the evolved sequenced clones (https://doi.org/10.6084/m9.figshare.14673177).

## Generation of capsule mutants

Isogenic capsule mutants were constructed by an in-frame deletion of *wcaJ* by allelic exchange as reported previously [36]. Deletion mutants were first verified by Sanger, and Illumina sequencing revealed that there were no off-target mutations.

## Conjugation assay

**Construction of pMEG-Mob plasmid.** A mobilizable plasmid named pMEG-Mob was built by assembling the region containing the origin of transfer of the pKNG101 plasmid [83] and the region containing the origin of replication, kanamycin resistance cassette, and IPTG-inducible *cfp* from the pZE12:CFP plasmid [84] (see Table C in S1 Text, and plasmid map, S8 Fig). We amplified both fragments of interest by PCR with the Q5 high fidelity DNA polymerase from New England Biolabs (NEB), with primers adapted for Gibson assembly designed with Snapgene, and used the NEB HiFi Builder mix following the manufacturer's instructions to assemble the 2 fragments. The assembly product was electroporated into electro-competent *E. coli* DH5α strain. KmR colonies were isolated, and correct assembly was checked by PCR. Cloned pMEG-Mob plasmid was extracted using the QIAprep Spin Miniprep Kit, and electroporation into the donor strain *E. coli* MFD λ-pir strain [85]. The primers used to generate pMEG-Mob are listed in Table D in S1 Text.

**Conjugation assay.** Recipient strains of *Klebsiella* spp. were diluted at 1:100 from a LB overnight into fresh LB in a final volume of 3 mL. Donor strain *E. coli* MFD λ-pir strain (diaminopimelic acid (DAP) auxotroph), which is carrying the pMEG-Mob plasmid, exhibited slower growth than *Klebsiella* strains and was diluted at 1:50 from an overnight into fresh LB + DAP (0.3 mM) + Kanamycin (50 μg/ml). Cells were allowed to grow at 37˚C until late exponential growth phase (Optical density; OD of 0.9 to 1) and adjusted to an OD of 0.9. The cultures were then washed twice in LB and mixed at a 1:1 donor–recipient ratio. Donor–recipient mixes were then centrifuged for 5 minutes at 13,000 rpm, resuspended in 25-μL LB+DAP, and deposited on a MF-Millipore Membrane Filter (0.45 μm pore size) on nonselective LB+DAP plates. The mixes were allowed to dry for 5 minutes with the lid open, and then incubated at 37˚C. After 1 hour, membranes were resuspended in 1-mL phosphate-buffered saline (PBS) and thoroughly vortexed. Serial dilutions were plated on selective (LB+Km) and non-selective (LB+DAP) plates to quantify the number of transconjugants (T) and the total

number of cells (N). The conjugation efficiency was computed with the following:

$$\text{Conjugation efficiency} = \frac{T}{N}$$

This simple method is relevant in our experimental setup because the plasmid can only be transferred from the donor strain to the recipient strain, and the duration of the experiment only allowed for minimal growth [86]. The lack of the conjugative machinery of MPF type I (the MPF type of RK2) within the plasmid and in the recipient strains prevents the transfer across recipient strains (see "Analysis of conjugative systems").

### Data analysis

All the data analyses were performed with R version 3.6 and Rstudio version 1.2. We used the packages ape v5.3 [87], phangorn v2.5.5 [88], and treeio v1.10 [89] for the phylogenetic analyses. Statistical tests were performed with the base package stats. For data frame manipulations and simulations, we also used dplyr v0.8.3 along with the tidyverse packages [90] and data. table v1.12.8.

### Supporting information

**S1 Fig. Comparison of 2 capsular loci.** Two CLTs (KL112 and KL24) involved in CLT swap, with the essential genes for capsule expression colored in blue. Gray tracks correspond to the sequence identity (computed using blastn) above 90% (see scale) to indicate highly similar homologs (liable to recombine). CLT, capsular locus type.
(TIFF)

**S2 Fig. Similarity between swapped CLT and other CLT. (A)** Comparison of sugar composition similarity (Jaccard similarity) between swapped vs. others CLTs. **(B)** Comparison of genetic similarity (wGRR) between swapped vs. others CLTs. The *p*-value displayed is for the 2-sample Wilcoxon test (https://doi.org/10.6084/m9.figshare.14673180). CLT, capsular locus type; wGRR, gene repertoire relatedness weighted by sequence identity.
(TIFF)

**S3 Fig. Phylogenetic distribution of the inactivated capsular loci.** The blue dots represent the putative inactivated capsules, which have at least 1 essential gene for capsule production pseudogenized or deleted (https://doi.org/10.6084/m9.figshare.14673156).
(TIFF)

**S4 Fig. Controls for the inactivated capsule gene analysis. (A)** Linear regression between the inactivation frequency normalized by average gene length and the rank of each gene in the biosynthesis pathway ($p = 0.005$, $R^2 = 0.7$) (https://doi.org/10.6084/m9.figshare.14673183). **(B)** Number of SSR in the core capsule genes in the Kaptive reference database (https://doi.org/10.6084/m9.figshare.14673174). **(C)** Genetic diversity of core capsule genes within the Kaptive reference database, represented by the percent of identity of all pairwise alignments of the proteins from different reference capsule loci (https://doi.org/10.6084/m9.figshare.14673192). SSR, simple sequence repeats.
(TIFF)

**S5 Fig. Distribution of the similarity measured by wGRR between pairs of conjugative systems (A) and between pairs of prophages (B) for wGRR >0.** The arrows represent the threshold (wGRR >0.99) set for clustering into families of highly similar elements. Since we performed transitive clustering to build the families, some elements belonging to the same

families have wGRR <0.99. We annotated the distribution of conjugation systems belonging to the same MPF type, which shows that systems of the same MPF are very similar but are below the selected threshold for clustering (https://doi.org/10.6084/m9.figshare.14673144 and https://doi.org/10.6084/m9.figshare.14673186). MPF, mating pair formation; wGRR, gene repertoire relatedness weighted by sequence identity.
(TIFF)

**S6 Fig. Changes in capsule state and branch length.** The capsule state changes among branches of the species tree are represented on the *x* axis, and the branch length is represented on the *y* axis in substitution per site. Individual points represent the mean for each group, and the bars represent the standard error. The *p*-values for *the t* test are represented on top of each comparisons (https://doi.org/10.6084/m9.figshare.14673159). We also performed a 2-sample Wilcoxon test to compare the medians ("Others" vs. "inactivation": $p < 0.0001$; "Others" vs. "Swap": $p < 0.0001$).
(TIFF)

**S7 Fig. Distribution of conjugation system MPF types.** Conjugation systems are classified in 2 categories according to their genomic location, which was predicted with the mlplasmids classifier. The MPF was predicted with the CONJscan module of MacSyfinder. Absolute number of systems are displayed for each category (https://doi.org/10.6084/m9.figshare.14673189). MPF, mating pair formation.
(TIFF)

**S8 Fig. pMEG-Mob plasmid genetic map.** pMEG-Mob was constructed by Gibson assembly from plasmids pKNG101 and pZE12. It encodes a colE1/pUC origin of replication (high copy number), a selectable marker (Kanamycin resistance cassette, green), the mobilizable region of pKNG101 which is composed of the origin of transfer of RK2 and 2 genes involved in conjugation (*traJ* and *traK*), a counter selectable marker (*sacB*), and an inducible CFP gene (IPTG induction). pMEG-Mob can only be mobilized in *trans* and thus can only be transferred from a strain expressing the RK2 conjugative machinery, which is absent from the panel of strains we used as recipients.
(TIFF)

**S1 Text. Supporting tables with references.** Supporting information containing detailed list of essential capsule genes, strains, plasmids, and primers used in this study.
(PDF)

## Acknowledgments

We thank Rafał Mostowy, Jorge Moura de Sousa, Nienke Buddlelmeijer, Olivier Tenaillon, Marie Touchon, and other lab members for fruitful discussions. We thank Sylvain Brisse for providing us with *Klebsiella* strains. We thank Christiane Forestier and Damien Balestrino for providing the pKNG101 plasmid and Jean-Marc Ghigo and Christophe Beloin for the gift of pZE12::CFP used to construct pMEG-Mob and *E. coli* MFD λ-pir.

## Author Contributions

**Conceptualization:** Matthieu Haudiquet, Olaya Rendueles, Eduardo P. C. Rocha.

**Data curation:** Matthieu Haudiquet, Olaya Rendueles, Eduardo P. C. Rocha.

**Formal analysis:** Matthieu Haudiquet.

**Funding acquisition:** Olaya Rendueles, Eduardo P. C. Rocha.

**Investigation:** Matthieu Haudiquet, Olaya Rendueles, Eduardo P. C. Rocha.

**Methodology:** Matthieu Haudiquet.

**Project administration:** Olaya Rendueles, Eduardo P. C. Rocha.

**Resources:** Matthieu Haudiquet, Amandine Buffet, Olaya Rendueles, Eduardo P. C. Rocha.

**Software:** Matthieu Haudiquet, Eduardo P. C. Rocha.

**Supervision:** Olaya Rendueles, Eduardo P. C. Rocha.

**Validation:** Matthieu Haudiquet, Olaya Rendueles, Eduardo P. C. Rocha.

**Visualization:** Matthieu Haudiquet.

**Writing – original draft:** Matthieu Haudiquet, Olaya Rendueles, Eduardo P. C. Rocha.

**Writing – review & editing:** Matthieu Haudiquet, Olaya Rendueles, Eduardo P. C. Rocha.

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
