## [Editor Report · Decision Letter 0]

22 Jan 2021

Dear Dr. Haudiquet, 

Thank you for submitting your manuscript entitled "Interplay between the cell envelope and mobile genetic elements shapes gene flow in populations of a nosocomial pathogen." for consideration as a Research Article by PLOS Biology.

Your manuscript has now been evaluated by the PLOS Biology editorial staff, as well as by an academic editor with relevant expertise, and I am writing to let you know that we would like to send your submission out for external peer review.

Please re-submit your manuscript within two working days, i.e. by Jan 24 2021 11:59PM.

Kind regards,

Paula

---

Associate Editor

PLOS Biology

---

## [Decision Letter · Decision Letter 1]

12 Mar 2021

Dear Dr. Haudiquet,

Thank you very much for submitting your manuscript "Interplay between the cell envelope and mobile genetic elements shapes gene flow in populations of a nosocomial pathogen." for consideration as a Research Article at PLOS Biology. Your manuscript has been evaluated by the PLOS Biology editors, an Academic Editor with relevant expertise, and by several independent reviewers.

In light of the reviews (below), we are pleased to offer you the opportunity to address the comments from the reviewers in a revised version that we anticipate should not take you very long. We will then assess your revised manuscript and your response to the reviewers' comments and we may consult the reviewers again.

You will see that the reviewers think that this is an interesting study but they all raise some issues that need to be solved before we can consider the manuscript further. In particular, reviewer #1 asks about evidence that capsule inactivation evolves in a similar manner under phage pressure, and about evidence for depolymerase gene exchange between different phages, in a reciprocal coevolutionary manner to capsule switches. Reviewer #2 and #3 want you to modify text and figures for clarity, and want you to add some points to the discussion. Reviewer #2 says that you could look at whether the galF-ugd locus is more prone to engage in recombination, suggests to add a phylogenetic tree indicating the branches with inactivated capsules, says that you should also take into account the size of the mutational target and asks whether you see any repeatability in the type of mutations isolated in the evolution experiment. This reviewer also asks whether you have checked if other plasmids carried by the strains are able to mobilize in trans the pMEG-Mob, and you should indicate whether you included appropriate controls in the conjugation experiments. Reviewer #3 says that you need to clarify cut-offs for deciding similarity in capsules, that you have to provide information about which strains were selected for evolution experiments and why, and have to explain what is the conjugation efficiency and show the strain serotypes. This reviewer also asks whether the passaged strains harbour other mutations and asks you to provide the changes vs the parental strain, and thinks that the data with longer branches should be excluded from the main results section. Please, address all the reviewers' comments. 

We expect to receive your revised manuscript within 1 month. Please, let us know if you need more time. 

**IMPORTANT - SUBMITTING YOUR REVISION**

*Resubmission Checklist*

*Published Peer Review*

*PLOS Data Policy*

*Blot and Gel Data Policy*

Sincerely,

Paula 

---

Associate Editor,

pjaureguionieva@plos.org,

PLOS Biology

REVIEWS:

Reviewer #1: Ecology, evolution and horizontal gene transfer in bacterial communities.

Reviewer #2: Plasmid dynamics in bacterial populations and bacterial evolution.

Reviewer #3: Bacterial plasmids for virulence and/or resistance.

Reviewer #1: In this manuscript, Haudiquet and colleagues bioinformatically analyse genome sequencing data to investigate the relationship between horizontal gene transfer (HGT) and capsule serotypes in the important opportunistic pathogen Klebsiella pneumoniae. They find an increased incidence of HGT within, rather than between, predicted serotypes. This they ascribe to transduction as a consequence of phage activity, which is often serotype-specific. They then investigate serotype swaps, which occur by recombination, probably with a brief intermediate uncapsulated state, which they experimentally show emerges readily in the laboratory by mutation. Capsule inactivation is associated with an excess of HGT, both by bioinformatic analyses, and increased conjugation efficiency in the lab, which can result in re-acquisition of a capsule. Together this analysis shows the state of flux in capsule presence, and how this process might structure gene exchange more broadly.

Overall I really enjoyed reading this paper — not only did it draw a dynamic and compelling (co)-evolutionary picture from comprehensive and sophisticated bioinformatic analyses, but it provided valuable experimental validation for several of the key mechanisms posited. As Klebsiella is a major clinical concern, I anticipate that the findings extend beyond evolutionary microbiology and will have broader relevance to genomic epidemiology. I have only a few comments and suggestions for the authors to consider in any revised submission.

- The proposed evolutionary process depends on phage predation to select for loss of the capsule, but the lab-evolved uncapsulated strains were evolved in LB media (and so were presumably under selection for capsule loss due to the metabolic cost of production). What evidence is there that capsule inactivation evolves in a similar manner under phage pressure?

- Line 137 looks for similarities between prophages of strains with similar capsule compositions and finds a weak proportionality. Would this relationship be strengthened by looking specifically at similarities between the depolymerase genes of those phages, rather than the phages as a whole? What evidence is there for depolymerase gene exchange between different phages, in a reciprocal coevolutionary manner to capsule switches?

Minor comments and suggestions:

- Figure 1E shows a summary of the means calculated across millions of points. Error bars or the full data should be presented (in supplementary information if not in the main figure). 

- Line 85 states that the capsule *needs* MGEs to vary by HGT. The authors should explain here why transformation is not considered a viable route.

- The sentence line 87-88 is not clear ('its'). 

- Somewhere in the paragraph starting line 317, the authors should make clear that capsule inactivation results in higher rates of gene *acquisition* by conjugation (rather than gene donation by conjugation). Does capsule presence affect outward conjugation?

- Figure 6 — state that conjugation efficiency is T/N in the legend.

- Figure 6. What is the known fitness benefit/cost of the non-capsulated mutants? This is unlikely to have a major effect on the outcome given the short period provided for conjugation, so perhaps the authors could give readers peace-of-mind by mentioning the short conjugation period either in the text or in the legend. 

- Figure S3. The panels are the wrong way round.

- Please check data availability box (accession numbers still read 'XXX'). 

Reviewer #2: In this work, Haudiquet et al. nicely explore the complex relationship between the capsule and HGT in Klebsiella pneumoniae (kpn). Through a combination of experiments and bioinformatic analyses, they convincingly show that phages typically infect kpn strains with same capsule type, whereas conjugative elements spread regardless of capsule type. Phages also drive capsule inactivation, which in turn increases the overall acquisition of conjugative elements, as the capsule hampers conjugative DNA transfer. Overall, the manuscript is well-written, the analyses are well-thought and statistically supported, the results are important and novel and will be of great interest to the broad audience of PLoS Biology. I only have some minor comments regarding data presentation and some experimental issues. 

- I found the introduction part describing capsule loci and the capsular locus types a little bit confusing (L57-62). The authors might want to include here the notion that Kpn capsule is encoded in the galF-ugt locus, and improve CLT definition to state what are the differences among distinct CLTs. In addition, thorough the manuscript the authors refer to CLT and serotype interchangeably. If both terms are really the same, I would suggest to stick to one of them for the sake of simplicity. 

- Figure 1A legend. The authors should state that the tree is a core genome tree of Kpn. 

- L109. I think the authors mistakenly refer to figure 1C instead of 1D?

- L120. The authors should add one or two sentences to explain a bit more wGRR, as it is a central parameter for their analyses. How should a non-expert interpret this parameter?

- Figure 1E. Why the data plotted only covers a small range of prophage wGRR (0.08-0.105), whereas in figure 1C it ranges from 0.2-1? Also, there seems to be some small dots on the plot that are not represented on the scale. 

- Figure 3A. Perhaps the figure y-axis could be changed to show the frequency instead of the absolute count, to better relate with the text (which states percentage of loci).

- L186. Although it is not essential for the main message of the manuscript, the authors could look at whether the galF-ugd locus is more prone to engage in recombination (i.e. hyper-recombinogenic) in comparison to other parts of the genome or the overall recombination rate of the species.

- L209. I would suggest the authors to show a phylogenetic tree indicating the branches with inactivated capsules as a supplementary figure, to better represent the idea that there's no phylogenetic signal.

- L229. Other interpretations are also possible. For instance, it may well happen that mutations in the first steps are more likely to occur (i.e. highly variable locus, homopolymeric tracts, insertion sequences). Although it is highly unlikely to affect the results, the size of the mutational target should also be taken into account. In addition, Do the authors see any repeatability in the type of mutations isolated in the evolution experiment? I do believe that the authors should provide a supp. table indicating which specific mutations were found on the evolution experiment. 

- L707-710. Most of the strains carry other plasmids apart from the pMEG-Mob (e.g. strain 342 has 6 plasmids). Have the authors checked that none of these plasmids are able to mobilize in trans the pMEG-Mob? Were appropriate controls included in the conjugation experiments? If so, it should be indicated. 

- I found really interesting that the main determinant of capsule swap is chemical composition and not genome similarity, which is what one would expect for recombining DNA fragments. The authors could discuss this further. 

Reviewer #3: This is an interesting paper which provides some evidence that strains of Klebsiella pneumoniae (Kpn) pass thhrough a stage of non-capsulation that might make them more susceptibe to HGT, especially by conjugative plasmids. Conjugation is likely to mediate examples of capsule switching they observe in their genomic data, which will alter the propensity of strains to be infected by capsule-recognising phages. Clearly the authors present data based mainly on bioinformatic analysis rather than experimentation (excpet for some conjugation experiments) so they need to be cautious drawing direct caasal links between sequence data and what happens during the natural lifecycle of Kpn. 

Specific comments

Data Fig 1 C refers to prophages while the text doesn't (line 107, We found significantly more genes acquired (co-gained) in parallel by different isolates having the same CLT than expected by simulations assuming random distribution in the phylogeny (1.95x, Z-test p<0.0001, Figure 1C), and it is not clear where these data are in the Figure. 

The authors should be claer about what their cut-offs are for deciding which capsules are similar and which are different. This is not provided in the results or materials and methods. 

136 The genomes with these CLTs, 59% of the total, show a weak but significant proportionality between prophage similarity and the number of similar residues in their host capsules(Figure 1E).. 

Please provide details

Please provide information about which strains did you select for evolution experiments and why? What are their capsule groups/phylogeny etc and did you recapitulate any mutations found in the epidemiologic collection… 

L340 Explain what the Conjugation efficiency is here please: Show the strain serotypes here please. 

These are passaged strains so might harbor other mutations. Is this the case? You have Illumina sequencing data. The changes vs the parental strains should be given as supllementary data. 

The branch length is much longer in strains with capsule swap (Fig S4) so I do not think that it is valid to include this group of strains in Fig 4 A and B. This might explain why there is evidence of more gene gain when there has been a swap rather than when there is capsule inactivation. Furthermore, are the events leading to swaps themselves discounted from these analyses? Although authors acknowledge this on line 268, the branches where capsules were swapped are 2.7 times longer than the others, precluding strong conclusions (Figure S4). So these data should be exclude from the main resultys section.

Also it is important to include in the discussion of the paper that it is possible that changes leading to capsule inactivation could arise during passage in vitro before sequencing and necessarily occur not in natural populations. 

98 How was this done? Around 92% of the isolates could be classed with good confidence-level.

143 Klebsiella quasipneumoniaesubsp. similipneumoniae (Kqs) explain please here? 

161 expected from previous works 

165 implicates. Better suggests

216 which strains did you select for evolution experiments and why? What are their capsule groups/phylogeny etc and did you recapitulate any mutations found in the epidemiologic collection… 

262 acquired in the branches present (not acquired)… 

292: conjugative systems

320 in the absence of published data on the frequency of conjugation in function of the presence of a capsule. Please check this sentence.

I am struggling with Figure 5 see what has been calculated. Please can you add further text to explain. 

364 more than wild type rather than other… 

379 There are multi-drug resistant hyper-virulent clones. Do you see evidence of capsule switches in these strains, which would be consistent with them experiencing a period of loss of capsule (when they could acquire the resistance plasmid) followed by restoration of capsule expression by conjugation. 

Figure 7 could be simplified. 

I do not think that both panels are necessary: 

Fewer bacteria in panel A would be less confusing… just use bacteria which are subject to the relevant steps. And in B, please colour the cap locus with the appropriate colour to highlight what is being transferred during conjugation…

COMMENTS FROM THE ACADEMIC EDITOR:

In addition to the points raised by the reviewers, the authors may want to clarify that variation in cell density of WT and non-capsulated versions of the strains used in the conjugation assays (Fig. 6) cannot explain the higher conjugation efficiency of non-capsulated strains. Cell density possibly affects mating opportunities in their filter matings beyond what can be controlled for by the ratio of transconjugants to total cell number. A statement that cell densities of WT and non-capsulated strains were not systematically different would suffice.

---

## [Editor Report · Decision Letter 2]

22 Apr 2021

Dear Dr. Haudiquet,

Thank you for submitting your revised Research Article entitled "Interplay between the cell envelope and mobile genetic elements shapes gene flow in populations of a nosocomial pathogen." for publication in PLOS Biology. I have now discussed with the Academic Editor. We will probably accept this manuscript for publication, provided you satisfactorily address the following data and other policy-related requests.

We recommend a change in the title to include the pathogen: "Interplay between the cell envelope and mobile genetic elements shapes gene flow in populations of the nosocomial pathogen Klebsiella pneumoniae."

DATA POLICY:

Regardless of the method selected, please ensure that you provide the individual numerical values that underlie the summary data displayed in the following figure panels as they are essential for readers to assess your analysis and to reproduce it: Figures 1B, 1C, 1D, 1E, 2A, 2C, 3A, 3B, 3C, 4A, 4B, 5, 6, Supplementary figures 2A, 2B, 4A, 4B, 4C, 5A, 5B, 6, 7.

**Please also ensure that figure legends in your manuscript include information on where the underlying data can be found, and ensure your supplemental data file/s has a legend.**

We expect to receive your revised manuscript within two weeks.

*Published Peer Review History*

*Early Version*

Sincerely,

Paula

---

Associate Editor,

pjaureguionieva@plos.org,

PLOS Biology

---

## [Editor Report · Decision Letter 3]

7 May 2021

Dear Dr. Haudiquet,

On behalf of my colleagues and the Academic Editor, Arjan de Visser, I am pleased to say that we can in principle offer to publish your Research Article "Interplay between the cell envelope and mobile genetic elements shapes gene flow in populations of the nosocomial pathogen Klebsiella pneumoniae." in PLOS Biology, provided you address any remaining formatting and reporting issues. These will be detailed in an email that will follow this letter and that you will usually receive within 2-3 business days, during which time no action is required from you. Please note that we will not be able to formally accept your manuscript and schedule it for publication until you have made the required changes.

PRESS

Thank you again for supporting Open Access publishing. We look forward to publishing your paper in PLOS Biology. 

Sincerely, 

Paula

---

Paula Jauregui, PhD 

Associate Editor 

PLOS Biology